# Characterization of Disinfection By-Products Originating from Residual Chlorine-Based Disinfectants in Drinking Water Sources

**DOI:** 10.3390/toxics12110808

**Published:** 2024-11-09

**Authors:** Dongmei Yang, Jiale Huang, Fenli Min, Huixian Zhong, Jialu Ling, Qun Kang, Zhaohua Li, Lilian Wen

**Affiliations:** 1College of Resource and Environmental Science, Hubei University, Wuhan 430062, China; 202321108012165@stu.hubu.edu.cn (D.Y.); 15972407666@163.com (J.H.); 202221108012325@stu.hubu.edu.cn (H.Z.); 202221108012330@stu.hubu.edu.cn (J.L.); kangqun@hubu.edu.cn (Q.K.); zli@hubu.edu.cn (Z.L.); 2Hubei Key Laboratory of Environmental and Health Effects of Persistent Toxic Substances, School of Environment and Health, Jianghan University, Wuhan 430056, China; minfenli2024@jhun.edu.cn

**Keywords:** drinking water source, disinfection by-products, chlorine-based disinfection, correlation analysis, chlorine content

## Abstract

In this study, samples from the Yangtze River, Han River, and Liangzi Lake in Wuhan City were utilized to characterize the formation of disinfection by-products (DBPs) from chlorine-based disinfection residues in drinking water sources. The results indicated that the main DBPs in drinking water sources were trichloromethane (TCM) and trichloroacetic acid (TCAA). The generation of DBPs was significantly positively correlated with oxidative substances, aromatic compounds, pH, and ammonia nitrogen (NH_3_-N) content in the water. The concentration of TCAA increased from 0 to 2.45 ± 0.31 mg/L when the reaction time increased to 72 h. As the NaClO concentration increased from 5 mg/L to 15 mg/L, the concentrations of TCAA, TBM, and DCAN increased from 2.03 ± 0.04 mg/L, 0 mg/L, and 0 mg/L to 2.49 ± 0.34 mg/L, 0.21 ± 0.07 mg/L, and 0.10 ± 0.04 mg/L before decreasing to 1.75 ± 0.19 mg/L, 0.17 ± 0.07 mg/L, and 0.04 ± 0.05 mg/L, respectively. The orthogonal experimental results showed that Br^−^, NH_3_-N, and pH all had significant influences on the TCM generation, whereas temperature affected the formation of TCAA in the Han River. This work reveals the factors influencing the generation of DBPs from chlorine-based disinfection residues, offering a prevention and control method for DBPs in drinking water sources from a theoretical perspective.

## 1. Introduction

The COVID-19 pandemic ravaging the world has posed severe challenges to the protection of drinking water sources, leading to widespread attention from society to the maintenance of public health drinking water. During the epidemic period, Hubei Province, especially Wuhan City, used a huge amount of chlorine-based disinfectants, with over 5000 tons in January–March 2020, far exceeding the use of other disinfectants. A large number of chlorine-based disinfection residues may have reached drinking water sources via urban wastewater treatment plants, rainwater erosion, and surface runoff, thus endangering the ecological environment and public health of drinking water sources [1,2]. Residual chlorine is an important water quality parameter of chlorine disinfection, and it has been required to be added as a monitoring index of drinking water sources. The maximum level of residual chlorine is 0.03 mg/L in China’s drinking water sources (Standards for drinking water quality (GB 5749-2022). 2023)[3]. The U.S. Environmental Protection Agency (EPA) stipulated that the residual chlorine value in natural water bodies should be less than 0.019 mg/L, and the European Union mandated that the residual chlorine value in freshwater bodies should be less than 0.005 mg/L [4].The potential impact of increased use of chlorine-containing disinfectants on drinking water safety and public health may have increased concentrations of DBPs in water sources, and further research is needed to assess and mitigate the risk. Considering the special conditions during the epidemic period, we assessed the change in DBPs by collecting and analyzing water samples during the epidemic period. We also studied the impact of high-concentration chlorine disinfectants on the formation of DBPs through simulation experiments, evaluated the impact of increased use of chlorine-containing disinfectants on the formation of DBPs under epidemic conditions, and explored strategies to reduce the formation of DBPs. And provide a scientific basis for future public health emergencies.

Residual chlorine will react with halides and natural organic substances to produce disinfection by-products (DBPs) that are harmful to human health during the disinfection and water distribution processes. Currently, at least 700 types of DBPs have been identified, the most prevalent of which are haloacetic acids (HAAs), halonitriles (HANs), and trihalomethanes (THMs) [5]. In 2023, dichloromethane (DCM) and trichloromethane (TCM) were included in the priority control list of the emerging pollutants in China. The extensive use of chlorine-containing disinfectants may lead to the fluctuation of DBPs composition and content in drinking water sources, which will affect the effluent water quality of water supply plants. Therefore, it is essential to characterize the formation of DBPs originating from chlorine-based disinfection residue in drinking water sources, thus guaranteeing the safety of public drinking water.

The type and dosage of disinfectants influenced the formation of DBPs. When the disinfectant was Cl_2_, the prevalent DBPs were THMs and HAAs [6], while THMs-type DBPs were produced by ClO_2_ [7]. As the dosage of chlorine increased, more DBPs were generated by substitution reactions. Research revealed that when the Cl_2_/DOC value of the sample increased from 0.5 to 3, the total abundance of Cl-DBPs increased from 5 × 10^6^ to 7 × 10^6^ [8]. Furthermore, the generation of DBPs was also affected by the level of bromide, reaction duration, and temperature [9,10,11]. Higher bromide concentrations and longer reaction times increased the yield of THMs, HAAs, and Halonitromethanes (HNMs) [12]. In the processes of chlorination and chloramination, higher temperatures also led to an increase in the generation of DBPs [13]. However, the impact of different environmental factors on the generation and transformation of DBPs has not been studied systematically, nor has the interaction between these factors. We focused on typical DBPs and explored their generation and transformation mechanisms. By identifying and controlling the formation of DBPs, negative impacts on aquatic ecosystems can be reduced, biodiversity can be protected, and the natural balance of water bodies can be maintained. The provided data and analysis illustrate the potential risks of DBPs to human health, and we discuss specific strategies to improve public health by reducing exposure to DBPs in order to provide a theoretical basis for future research on the impact of various environmental factors on the generation of DBPs.

In this study, a series of experiments was carried out using drinking water samples from the Yangtze River, Han River, and Liangzi Lake in Wuhan City, and four typical DBPs, including trichloromethane (TCM), trichloroacetic acid (TCAA), tribromomethane (TBM), and dichloroacetonitrile (DCAN), were measured to reveal the formation and transformation of DBPs in natural water bodies.

## 2. Materials and Methods

### 2.1. Chemicals

Sodium hypochlorite (NaClO, 5.2%), hydrochloric acid (HCl, 36–38%), sulfuric acid (H_2_SO_4_, 95–98%), sodium hydroxide (NaOH, 96%), potassium nitrate (KNO_3_), monosodium phosphate (NaH_2_PO_4_), disodium phosphate (Na_2_HPO_4_), potassium bromide (KBr), and ammonium chloride (NH_4_Cl) were purchased from Sinopharm Chemical Reagent Co., Ltd. in Shanghai, China. Potassium persulfate (K_2_S_2_O_8_) and N-1-Naphthylethylenediamine dihydrochloride (CHN₂·2HCl) were purchased from Shanghai Macklin Biochemical Technology Co., Ltd. in Shanghai, China. Potassium permanganate (KMnO_4_) was purchased from Aladdin Reagent (Shanghai) Co., Ltd. in Shanghai, China, and all the reagents were chemical analysis-grade.

### 2.2. The Formation of DBPs in Raw Water from Different Water Sources

Liangzi Lake is the largest freshwater lake in Wuhan City. The Han River and Yangtze River are the two largest rivers flowing through Wuhan City. Therefore, the reason why we choose these three water sources is that these three rivers are closely related to the drinking water safety of Wuhan’s people. To investigate the impact of water quality characteristics and sampling methods on the formation of DBPs from different water sources, water samples were collected from the Zongguan site of the Han River, the Junshan site of the Yangtze River, and Liangzi Lake in Wuhan using both glass bottles and polyethylene bottles in October 2021. The depths of the sampling sites ranged from 10% to 25% water depth of the intake of the water supply plant. The sampling locations are illustrated in Figure 1.

The sampling bottles were cleaned three times with deionized water and then three times with water straight from the water source before sample collection. No air was left at the tops of the bottles during collection. The water samples were processed according to the pretreatment methods described in GB/T 5750.2-2006 [14] “Standard Examination Methods for Drinking Water”. The samples were stored in Styrofoam containers filled with ice and treated with ascorbic acid to halt chlorination reactions. After that, samples were returned to the lab at low temperatures and kept in a 4 °C refrigerator. Glass bottle samples from Liangzi Lake, Han River, and Yangtze River were named LG, HG, and CG, respectively. And polyethylene bottle samples from Liangzi Lake, Han River, and Yangtze River were called LP, HP, and CP, respectively.

### 2.3. Factors Influencing the Formation of DBPs in Water Sources

Firstly, 50 mL amber glass bottles with screw caps were uniformly selected as the reaction vessels, and the reaction systems all had aqueous volumes of 25 mL.

To investigate the impact of reaction time on the species and concentrations of DBPs, the reaction solution was adjusted to 10 μmol/L Br^−^ and 10 mg/L effective chlorine by adding a certain amount of KBr (1 mg/L) and NaClO (1000 mg/L). The reaction mixture was then placed in a dark, constant-temperature incubator at 20 °C for 72 h. Samples were taken at 6, 12, 24, 36, 48, and 72 h, and the reaction was terminated with ascorbic acid at each time point. Each water source was subjected to two parallel experiments.

To study the impact of chlorine disinfectant dosage on the formation of DBPs, the chlorine concentrations were set as 5, 10, and 15 mg/L. The reaction mixtures were put in a constant-temperature incubator at 20 °C in the dark for 24 h. The reaction was terminated with ascorbic acid, and two replicates were performed for each treatment.

To explore the influence of the factors Br^−^, NH_3_-N, pH, and temperature on DBPs, an orthogonal table was used to arrange the experiments as shown in Table 1. The Br^−^ concentrations were set to 0.5, 1, and 1.5 mg/L, NH_3_-N concentrations were set to 1, 3, and 5 mg/L, and pH values were set to 6 ± 0.2, 7 ± 0.2, and 8 ± 0.2. Different constant-temperature incubators were utilized to adjust the temperatures to 15, 20, and 25 °C, with two parallel experiments conducted for each set.

### 2.4. Chemical Analyses

The DBPs determined in this experiment include TCM, TCAA, TBM, and DCAN. Sample pretreatment involved the use of liquid–liquid extraction to separate and enrich DBPs. Specifically, 10 to 20 mL of the water sample was placed in a centrifuge tube, and 5 mL of dichloromethane (DCM) (analytical grade) was added. After covering the tube, it was subjected to agitation at 100 rpm on a constant-temperature shaker for 5 min, followed by a 5 min settling period. Subsequently, 1 µL of the bottom liquid was extracted for gas chromatography detection, while 10 µL was used for liquid chromatography detection.

The Agilent 7890B gas chromatograph (GC) was employed for the detection of TCM, TBM, and DCAN. The gas chromatograph was equipped with a Flame Ionization Detector (FID) (Danbury, CT, USA) and utilized an Agilent Scientific capillary column (30 m × 0.25 mm, 0.25 µm, 5% phenyl methyl siloxane). Nitrogen (N_2_) was used as the carrier gas, with a flow rate maintained at 1 mL/min. The injection mode was split-less, and the column pressure was 9.08 Psi. The temperatures of the injector and detector were set at 260 °C and 150 °C, respectively. The oven temperature program involved an initial temperature of 60 °C held for 1 min, followed by a ramp at 12 °C/min to 200 °C, which was held for 1 min, and then a further ramp at 5 °C/min to 325 °C, held for 1 min.

Shimadzu LC2030 high-performance liquid chromatography (HPLC) (Kyoto, Japan) was selected to detect TCAA, choosing the Waters Xbridge C18 column (250 mm × 4.6 mm, 5 µm, from Milford, MA, USA) as the chromatographic column, with the column temperature setting at 40 °C. A 0.1 mol/L solution of ammonium dihydrogen phosphate (mobile phase A) and acetonitrile (mobile phase B) was used to perform gradient elution, with the pH and the flow rate adjusted to 3.0 and 1.2 mL/min, respectively. The elution conditions included an isocratic elution with mobile phase A at 80% and mobile phase B at 20% for the initial 15 min, followed by a linear gradient to 10% mobile phase A and 90% mobile phase B over 15 min, and then returned to the initial condition over the next 5 min. Detection was performed by a UV detector at a wavelength of 210 nm.

NH_3_-N and total nitrogen (TN) were detected by applying a Shimadzu UV-3600 UV-Vis spectrophotometer (Kyoto, Japan). The water sample’s pH was adjusted to 5–9 using NaOH or H_2_SO_4_, and suspended solids were removed. A cuvette was employed, and detection wavelengths were set at 220 nm and 275 nm for TN and 420 nm for NH_3_-N. For ultraviolet absorbance at 254 nm (UV_254_) detection, the sample was filtered, a cuvette was used again with deionized water as a reference, and the detection wavelength was set at 254 nm on the UV-Vis spectrophotometer.

The total organic carbon (TOC) was measured using the Shimadzu TOC-V total organic carbon analyzer (Kyoto, Japan). The water sample was acidified and filtered. The instrument was then preheated for 2 h and the total carbon combustion tube temperature was set at 900 °C, the inorganic carbon reaction tube temperature was set at 160 ± 5 °C, and the carrier gas flow rate was adjusted to 180 mL/min.

A Hach HQ40d multi-parameter water quality analyzer (Loveland, CO, USA) was utilized to detect electrical conductivity (EC), dissolved oxygen (DO), chlorine residue, chemical oxygen demand (COD), and dissolved organic carbon (DOC). Additionally, the Shanghai Leici PHS-2F pH meter was used to measure the pH value of the water sample.

### 2.5. Statistical Analyses

We used one-way analysis of variance (ANOVA) and multi-factor analysis of variance to analyze the water quality data from three different water sources, investigating the significant differences in DBPs due to distinct sampling methods and water sources, as well as the significant effects of factors such as Br^−^, NH_3_-N, pH, and temperature on DBPs. Principal component analysis (PCA) on EC, TN, pH, NH_4_^+^, TOCl, COD, UV_254_, DO, DOC, SUV_254_, and TCAA was performed to explore the reduced dimensions of parameters and sample clustering using Origin. We utilized the Spearman correlation coefficient to analyze the correlations between EC, TN, pH, NH_4_^+^, TOCl, COD, UV_254_, DO, DOC, SUV_254_, and TCAA and plotted a correlation heatmap to investigate the relationships among these indicators.

## 3. Results and Discussion

### 3.1. The Occurrence of DBPs in Different Water Sources

During the generation of DBPs, pH, EC, DO, COD, UV_254_, SUV_254_, and temperature may be directly related to the conversion and the rate of DBPs, while DO, TN, NH_4_^+^, and DOC indirectly indicate the precursors of DBPs. In order to examine the initial occurrence of DBPs in water sources, the pH, EC, DO, temperature, chlorine residue, TN, NH_4_^+^, COD, DOC, UV_254_, and SUV_254_ indicators of water samples taken from the Han River, Yangtze River, and Liangzi Lake were measured using instruments such as a thermometer, a portable pH meter (PHS-2F) (Leici, Shanghai, China), and a multiparameter water quality analyzer (HQ40d) (Hach, Loveland, CO, USA). To express the significant relationship between various water quality indicators and the generation of DBPs, analysis of variance (ANOVA) was carried out, and the results are shown in Table 2.

Different sampling methods had no significant impact on water quality indicators, but significant differences were observed from different water sources. The chlorine residue indicator’s *p*-value was only 0.1880 between the two sampling methods, which was lower than other indicators. This could be due to the reaction between residual chlorine and plastic bottles, which reduced the amount of residual chlorine in the polyethylene bottles. Consequently, the following studies were carried out in glass bottles to avoid influencing the formation of DBPs.

TCAA was present in water sources, except the Yangtze River, with 0.160 ± 0.003 mg/L in the Han River and 0.086 ± 0.006 mg/L in Liangzi Lake. The statistical analysis showed that there were significant differences (*p* = 0.000) in the formation of TCAA among the various water sources.

### 3.2. Correlation Analysis of DBPs and Water Quality Indicators

To further determine the primary sources of DBPs, a Spearman correlation analysis and principal component analysis were conducted on water quality parameters (pH, EC, TN, NH_4_^+^, TOCl, COD, UV_254_, DO, DOC, SUV_254_, TCAA). Figure 2a reflects the correlation between various indicators among different water sources, while Figure 2b illustrates the correlations and interactions between them. The explanatory power of the PC1 axis was 59.1%, and that of the PC2 axis was 33.1%, with the total contribution of the two main components reaching 92.2%. This implied that during the dimensionality reduction analysis, all relevant information was preserved, indicating that the parameters were able to effectively reflect the experimental results. Apart from TCAA, UV_254_ was commonly used to assess the aromaticity and the content of conjugated organic compounds in water quality [15], while DOC primarily consisted of natural organic matter, and their ratio relationship was indicated by SUV_254_, expressed as ${\mathrm{SUV}}_{254}=\frac{{\mathrm{UV}}_{254}}{\mathrm{DOC}}\times 100$. The value of SUV_254_ to a certain extent reflected the hydrophilicity, aromaticity, and content of the macromolecular organic matter of DOC [16]. Hence, UV_254_, SUV_254_, and DOC could serve as reference indicators for the precursors of DBPs.

The correlation analysis (as shown in Figure 2a) indicated a strong positive correlation between TCAA, COD, and UV_254_, with correlation coefficients of 0.87 and corresponding *p*-values of 0.028 and 0.014, suggesting that the generation of TCAA in water sources was significantly influenced by the oxidative substances and aromatic compounds in the water [17,18,19]. With correlation coefficients of 0.956 and 0.943, pH, TN, and SUV_254_ were found to have significant relationships when compared to the precursors of DBPs. This indicated that the pH and different nitrogenous organic compounds had a significant influence on the composition and properties of precursors of DBPs from various sources of drinking water [20,21,22,23]. The Han River source water was more likely to produce TCAA according to the main component analysis (shown in Figure 2b). The high correlation of TCAA with COD and UV_254_ suggested that the generation of TCAA was closely linked to the oxidative substances and aromatic compounds in the water, in alignment with the aforementioned findings.

Hence, the generation of DBPs from different water sources was significantly related to the presence of oxidative substances, aromatic compounds, water pH, and NH_4_^+^ content.

### 3.3. The Effects of Reaction Time on Generation of DBPs

To investigate the variation in the generation of DBPs with reaction time, the Br^−^ concentration in the solution was set at 10 μmol/L and the effective chlorine at 10 mg/L. The solution was then subjected to a dark reaction at 20 °C for 72 h in a constant-temperature incubator. During the reaction, water samples were collected at 6, 12, 24, 36, 48, and 72 h intervals to measure the content of DBPs, as illustrated in Figure 3.

The results indicated that TCM and TCAA were the main DBPs generated in the water sources of Liangzi Lake, Han River, and Yangtze River. Overall, the concentration of TCM increased significantly compared with the initial non-chlorinated state. This suggests that as the reaction time progressed, more trihalomethanes (THMs) formed, which aligns with previous studies showing that prolonged chlorination promotes THM generation. However, the lower TCM levels in the Han River compared to Liangzi Lake may indicate the presence of other factors limiting its formation, such as the water’s organic matter content or lower precursor availability. These factors may affect the generation of THMs and are worth further exploration in future research. The TCM generation in Liangzi Lake peaked at 1.90 mg/L after 72 h, while the TCM generation in the Han River peaked at 0.40 mg/L after 12 h and stabilized at 0.22 mg/L. After 48 h, the Yangtze River’s TCM generation peaked at 0.48 mg/L and subsequently settled at 0.39 mg/L. This phenomenon might be explained by enough reaction time causing the precursor levels of DBPs in the water to decrease, which in turn caused the Han and Yangtze Rivers to generate TCM at a slower rate. Liangzi Lake’s consumption might have been lower than other lakes due to its greater TOCl content, which directly contributed to the creation of TTHMs [24,25]. As the reaction time increased, the stored TOCl in Liangzi Lake water began to deplete, leading to the generation of new TCM precursors and thereby promoting TCM formation.

Moreover, the concentrations of TCAA in the water samples from all three sources initially increased, then gradually decreased and eventually stabilized. Liangzi Lake and Yangtze River peaked at 3.03 mg/L and 2.94 mg/L, respectively, after 12 h, and the Han River peaked at 2.94 mg/L after 24 h. The concentrations of TCAA in Liangzi Lake, Han River, and Yangtze River declined slightly as the reaction time increased and then stabilized at 2.17 mg/L, 2.52 mg/L, and 2.57 mg/L, respectively. The decrease in TCAA levels may be attributed to its poor stability in water at room temperature, leading to decomposition and the generation of TCM and tetrachloroethylene [26,27]. Additionally, Br^−^ may have caused a gradual transition of chlorinated DBPs to brominated DBPs, resulting in a decrease in TCAA levels over time.

TBM and DCAN were not detected at any levels during the experiment. This might be because of their low generation levels, which were below detection limits, or because they transformed into other THMs, HANs, etc. The poor stability of DCAN may have caused the high TOCl levels to inhibit its generation, leading to its transformation into other DBPs over time [28]. As the chlorination disinfection time increased, larger-molecular-weight brominated DBPs were initially formed, which then gradually decomposed into smaller-molecular-weight brominated DBPs [29].

In summary, during the first 12 h of chlorination disinfection, significant amounts of TCM and TCAA were generated. As the reaction time increased, the contents and types of precursors in water from different sources underwent different changes. During this time, the ratio of Br^−^ to DOC in water continuously increased, making it more susceptible to generating brominated DBPs [30]. Beyond 12 to 48 h, the subsequent generation or degradation rates decreased, and it was likely that further transformation into other DBPs would occur after 48 h.

### 3.4. The Effect of Different Concentrations of NaClO on the Generation of DBPs

During the disinfection process, it was essential to strictly control the dosage of chlorinated disinfectants to ensure the safety of the water supply. To investigate the impact of different concentrations of disinfectants on the generation of DBPs, the Br^−^ concentration in the solution was maintained at 10 μmol/L, and different concentrations of NaClO were added to achieve effective chlorine concentrations of 5, 10, and 15 mg/L. The solution was then subjected to a dark reaction at 20 °C for 24 h in a constant-temperature incubator. The experimental results are depicted in Figure 4.

Liangzi Lake was the primary source of TCM and TCAA generation; TCM, TCAA, TBM, and DCAN were formed in the other two water sources. In the investigation of the impact of reaction time on the generation of DBPs under the same conditions, no TBM and DCAN were detected, possibly due to experimental condition variations in temperature and pressure causing differences in reactions and rendering them undetectable. With increased chlorine dosing, a decreasing trend of TCM generation was observed in Liangzi Lake and the Yangtze River, while other detected DBPs from all water sources showed an initial increase followed by a decrease. Liangzi Lake, Han River, and the Yangtze River all showed maximum TCAA generation levels of 2.37 mg/L, 2.88 mg/L, and 2.23 mg/L at a chlorine dosage of 10 mg/L, while Han River showed a maximum TCM generation level of 0.40 mg/L. Additionally, at a chlorine dosage of 10 mg/L, TBM and DCAN began to appear and reached maximum levels in the Han River and the Yangtze River, with values of 0.26 mg/L and 0.13 mg/L for the Han River and 0.16 mg/L and 0.07 mg/L for the Yangtze River.

In conclusion, the generation of TCM in the Han and Yangtze Rivers progressively reduced as the chlorine dose increased, whereas the development of TBM tended to increase, indicating the availability of precursors that favor the generation of brominated DBPs. Higher chlorine dosages may facilitate the transformation of some chlorinated trihalomethanes into brominated trihalomethanes. Furthermore, the gradual decrease in DCAN in the Han River and its disappearance in the Yangtze River with increased chlorine dosage indicated potential degradation or transformation of DCAN into brominated acetonitrile. The subsequent decrease in the generation of DBPs may be attributed to partial transformation or volatilization [31,32].

### 3.5. Investigation of Formation of DBPs in Water Sources Under Various Environmental Factors

To investigate the impact of four factors, including Br^−^, NH_4_Cl, pH, and temperature, on the generation of DBPs, an orthogonal experiment was conducted as illustrated in Appendix A (Appendix A). The experiment ultimately detected the values of TCM and TCAA in water from various sources and did not detect TBM or DCAN. Subsequently, the magnitude of the impact of the four factors on DBPs was determined by analyzing the R values, with smaller concentrations of DBPs indicating a lesser impact and larger concentrations indicating a greater impact. A multifactor analysis of variance was utilized to examine the differential relationships between Br^−^, NH_4_Cl, pH, and temperature in different water sources with regard to TCM and TCAA. Taking the Han River as an example, the range R’ and significance analysis of TCM and TCAA are presented in Table 3, and the trend of DBPs content in the Han River under each experimental factor is illustrated in Appendix A.

In the Han River, the generation of TCM was significantly influenced by Br^−^, NH_4_Cl, and pH, while TCAA generation was primarily affected by temperature. The relative impact of the four factors on TCM generation was as follows: temperature < Br^−^ < pH < NH_4_Cl. The optimal combination of conditions for TCM generation in the Han River water source was determined to be Br^−^ 1.5 mg/L, NH_4_Cl 5 mg/L, pH 8, and a temperature of 15 °C. Conversely, the relative impact of the four factors on TCAA generation in the Han River water source was: NH_4_Cl < pH < Br^−^ < temperature. The optimal combination of conditions for TCAA generation in the Han River water source was determined to be Br^−^ 0.5 mg/L, NH_4_Cl 5 mg/L, pH 7, and temperature 25 °C. The influence of temperature on the generation of the two DBPs in the Han River water source exhibited a notable difference, indicating that temperature may have a certain impact on the composition of TCAA precursors in the Han River water source [33]. The contrasting impact of the factors on TCM and TCAA in the Han River water source suggested that under certain conditions, the water quality could inhibit TCM generation when TCAA was generated in large quantities, while this phenomenon did not include effective chlorine concentrations of 5, 10, and 15 mg/L. Conversely, when the TCAA content decreased due to changes in water quality conditions, TCM could be generated in substantial amounts. This was consistent with the earlier findings indicating changes in DBPs in the Han River water source influenced by the factor of reaction time.

## 4. Conclusions

The purpose of this study was to investigate the influences of different environmental factors on the generation and transformation of DBPs, as well as the interactions between these factors. Through a series of experiments and data analysis, the following conclusions could be drawn. The composition and characteristics of DBPs precursors were shown to be significantly correlated with NH_3_-N concentration, aromatic chemicals, oxidative substances, and water pH levels. Increased reaction time promoted the generation of DBPs. The different types of DBPs generally showed an initial increase followed by a declining trend with the increase in effective chlorine concentration. The generation of TCM showed significant variations with changes in Br^−^, NH_4_Cl, pH, and temperature, while TCAA may be correlated with other factors. There was evidence of partial decomposition or transformation of some DBPs in the later stages of the reaction. These findings demonstrate the significance of pH, TN, COD, and UV_254_ in relation to DBPs. They also show the effects of temperature, pH, Br^−^, NH_4_Cl, and reaction time on the formation of DBPs. This provides theoretical support for the control of DBPs in subsequent studies. 

Through this research, we should reduce chlorination time, control the dosage of chlorinated disinfectants, and select the optimal conditions for four environmental factors in actual water treatment practices. Later, we can further investigate the optimal concentrations of chlorinated disinfectants and four environmental factors for the purpose of reducing the generation of DBPs and study the transformation mechanism of organic acids and humic substances in the system through fluorescence PCR technology and other methods.

## Figures and Tables

**Figure 1 toxics-12-00808-f001:**
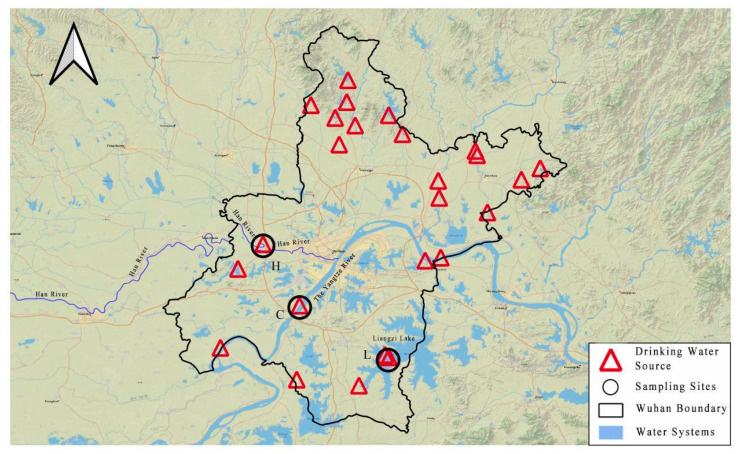
Distribution map of drinking water sources in Wuhan.

**Figure 2 toxics-12-00808-f002:**
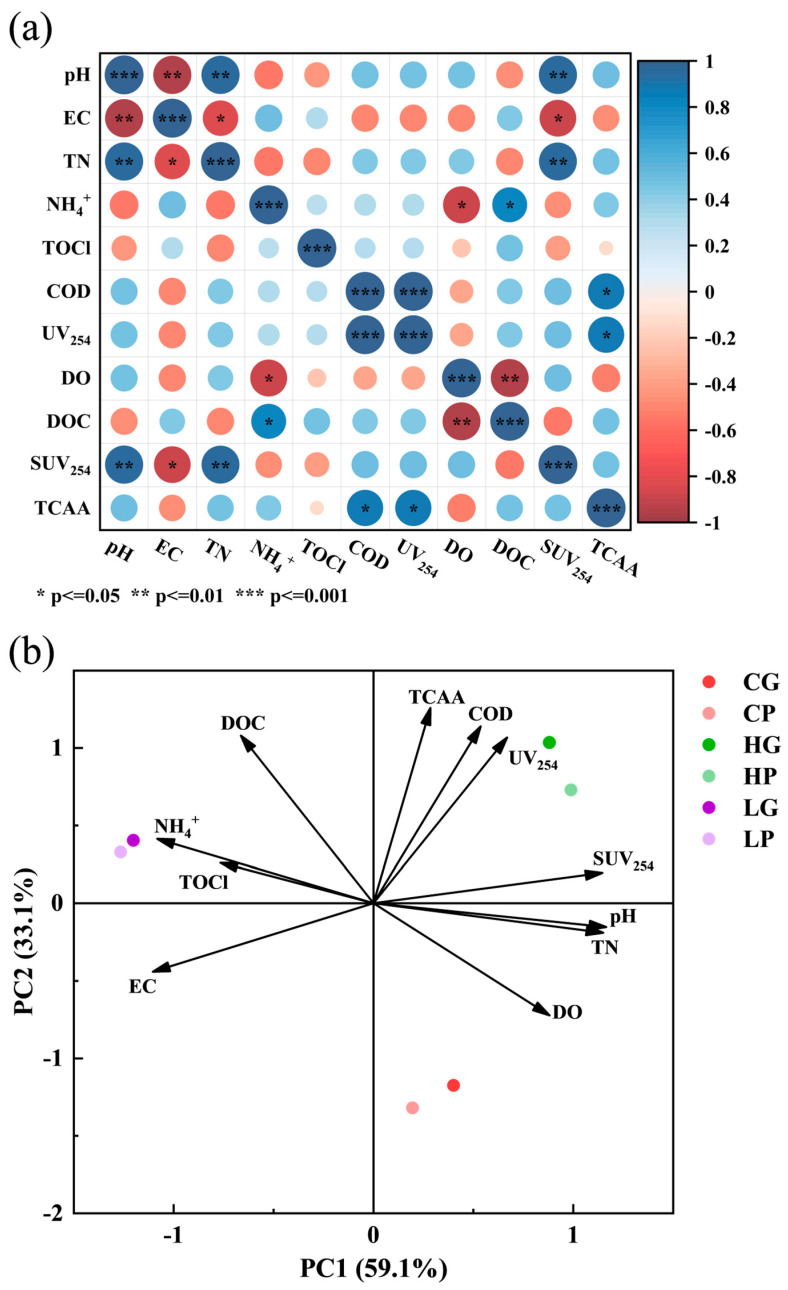
Correlation analysis (**a**) and principal component analysis (**b**) of disinfection by-products (DBPs) with water quality parameters. In Figure (**a**), blue and orange indicate positive and negative correlations between the two factors, respectively. The larger the circle, the stronger the correlation, while the smaller the circle, the weaker the correlation. TOCl is total organic chlorine, TCAA is trichloroacetic acid, and SUV_254_ is spectral UV absorption at 254 nm. In Figure (**b**), CG, HG, and LG mean samples in glass bottles from the Yangtze River, Han River, and the Liangzi Lake, respectively, while CP, HP, and LP mean samples in polyethylene bottles from the Yangtze River, Han River, and the Liangzi Lake, respectively. Variables with arrows pointing in the same direction have a positive correlation on principal components, while variables pointing in the opposite direction have a negative correlation. The length of the arrow represents the variance contribution of the variable, and the longer the length, the greater the explanatory power of the variable on the principal component.

**Figure 3 toxics-12-00808-f003:**
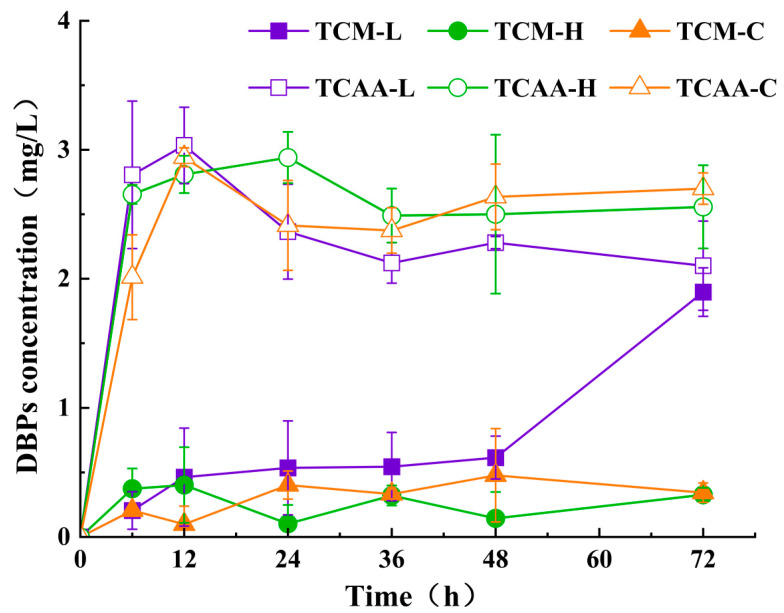
The time course of concentrations of DBPs under the condition of 10 mg/L chlorine and 10 μmol/L Br^−^ at 20 °C. TCM-L, TCM-H, and TCM-C mean trichloromethane (TCM) in Liangzi Lake, Han River, and Yangtze River, respectively, while TCAA-L, TCAA-H, and TCAA-C mean trichloroacetic acid (TCAA) in Liangzi Lake, Han River, and Yangtze River, respectively.

**Figure 4 toxics-12-00808-f004:**
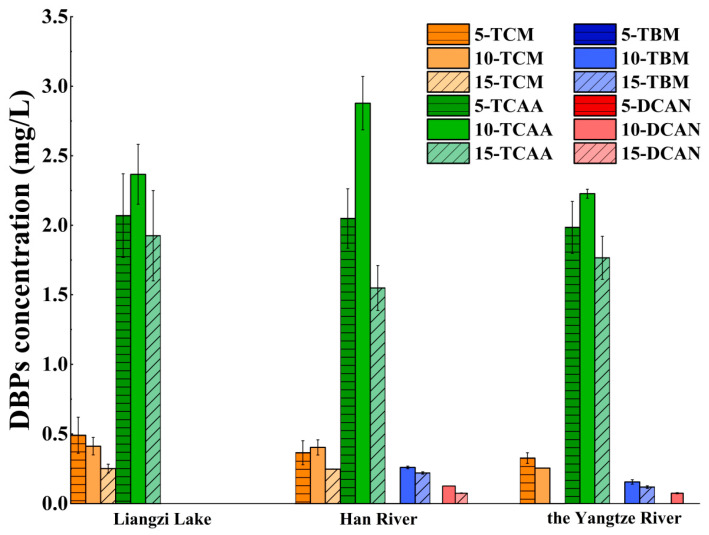
Effect of chlorine concentrations and water sources on the formation of disinfection by-products (DBPs) under the dark condition of 10 μmol/L Br^−^ at 20 °C for 24 h. 5-TCM, 10-TCM, and 15-TCM mean that the concentrations of trichloromethane (TCM) are 5 mg/L, 10 mg/L, and 15 mg/L respectively, while 5-TCAA, 10-TCAA, and 15-TCAA mean that the concentrations of trichloroacetic acid (TCAA) are 5 mg/L, 10 mg/L, and 15 mg/L respectively. 5-TBM, 10-TBM, and 15-TBM mean that the concentrations of tribromomethane (TBM) are 5 mg/L, 10 mg/L, and 15 mg/L respectively, while 5-DCAN, 10-DCAN, and 15-DCAN mean that the concentrations of dichloroacetonitrile (DCAN) are 5 mg/L, 10 mg/L, and 15 mg/L respectively.

**Table 1 toxics-12-00808-t001:** Orthogonal experimental design table.

Group	Br^−^ (mg/L)	NH_4_Cl (mg/L)	pH	Temperature (°C)
1	0.5	1	6	15
2	0.5	3	7	20
3	0.5	5	8	25
4	1	1	7	25
5	1	3	8	15
6	1	5	6	20
7	1.5	1	8	20
8	1.5	3	6	25
9	1.5	5	7	15

**Table 2 toxics-12-00808-t002:** Physical and chemical properties in water samples and the effect of sampling methods and water sources on the values of water quality indicators using one-way analysis of variance.

Water Quality Indicators	Sample Name						*p*-Value	
	LG	HG	CG	LP	HP	CP	A ^1^	B ^2^
EC	1025	756	916	1024	757	917	0.998	0.000 **
DO	9.800	10.200	10.600	9.500	10.100	10.300	0.523	0.045 *
TOCl	0.256	0.101	0.092	0.092	0.034	0.058	0.188	0.390
TN	0.581	1.151	1.101	0.546	1.323	1.144	0.851	0.005 **
NH_4_^+^	0.290	0.126	0.126	0.355	0.144	0.120	0.797	0.009 **
COD	16.700	17.900	16.100	16.300	17.400	15.800	0.603	0.021 *
DOC	4.210	3.940	3.140	4.220	3.720	3.170	0.899	0.003 **
UV _254_	0.175	0.224	0.167	0.169	0.216	0.151	0.723	0.008 **
SUV _254_	4.160	5.690	5.320	4.010	5.810	4.760	0.792	0.014 *
TCAA	0.082	0.158	0	0.090	0.162	0	0.954	0.000 **

^1^ Grouping according to different sampling methods; ^2^ grouping according to different water sources; * the results have a significant influence when *p* ≤ 0.05; ** the results have a significant influence when *p* ≤ 0.01.

**Table 3 toxics-12-00808-t003:** The range and significance analysis of concentrations of disinfection by-products (DBPs) under four environmental factors by analysis of variance in Han River.

DBPs	TCM		TCAA	
	Range	*p*	Range	*p*
Br^−^ (mg/L)	0.07	0.001 **	0.95	0.334
NH_4_Cl (mg/L)	0.25	0.000 **	0.49	0.761
pH	0.17	0.001 **	0.54	0.711
Temperature	0.06	0.11	1.97	0.044 *

* The results have a significant influence when *p* ≤ 0.05; ** The results have a significant influence when *p* ≤ 0.01.

## Data Availability

All data are included in the manuscript.

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
