# Peer review of "Characterization of Disinfection By-Products Originating from Residual Chlorine-Based Disinfectants in Drinking Water Sources"

_toxics, 2024, doi:10.3390/toxics12110808_

Round 1
Reviewer 1 Report
Comments and Suggestions for Authors
Reviewer comments and suggestions
Manuscript ID: toxics-3271126
With this manuscript, the authors investigated disinfection by-products (DBPs) that were formed when using chlorine-based disinfection in drinking water sources across three water bodies in Wuhan City, China: the Yangtze River, Han River, and Liangzi Lake. This type of study is important because it permits an understanding of what factors influence DBP formation from chlorine-based disinfection residues, to help prevent and control DBPs in drinking water sources.
In my opinion, the manuscript is suitable for publication in Toxics, after revision, because the importance and innovation, of the paper are not totally clearly demonstrated.
Some concrete comments would be as follows:
Introduction
- While the authors establish background context by discussing the COVID-19 pandemic's effects on chlorine-based disinfectant usage, the introduction would benefit from a stronger articulation of the research objectives and its contributions to the field to help readers understand the research's novelty.
- In this section, the authors should highlight how their research extends beyond existing studies and demonstrate the importance of DBPs research for both environmental protection and public health outcomes.
- Despite incorporating multiple DBP studies, the literature review's organization could be improved. The previous research should be presented in a more organized manner, leading logically to the current study's objectives.
- The justification for the chosen water sources (Yangtze River, Han River, Liangzi Lake) should be more explained.
Methodology
-The authors should better define acronyms and abbreviations upon first mention.
Results and Discussion
- The authors should enhance the results section by including more comparative analysis with existing research and ensuring proper integration of all Figures and Tables. Ensure all figures and tables are well-referenced and sufficiently explained during the text.
-Table 2 (Water Quality Indicators): The data is presented without explaining its relevance to DBP formation or how it supports this study.
- For example, Figure 3 (Reaction Time on DBPs Generation) - "The results indicated that TCM and TCAA were the main DBPs generated in the water sources of Liangzi Lake, Han River, and Yangtze River. Overall, the TCM content in the water samples from all three sources showed an increasing trend over time."
The reviewer suggests after stating the increase of TCM, the authors could add: "This suggests that as the reaction time progresses, more trihalomethanes (THMs) form, which aligns with previous studies showing that prolonged chlorination promotes THM generation. However, the lower TCM levels in the Han River compared to Liangzi Lake may indicate the presence of other factors limiting its formation, such as the water’s organic matter content or lower precursor availability ….."
- Section 3.2 (Correlation Analysis of DBPs and Water Quality Indicators): Why is this correlation important in the context of DBP formation and how is it tied to the study’s overall goal?
-To improve data interpretation, the authors should enhance the labeling and expand the captions of several figures, especially Figures 2 and 3. This would allow readers to understand the data without constantly referring to the main text for explanation.
Conclusion
-It provides a good summary of results, but the authors should outline how their findings could influence water treatment practices and identify areas requiring further investigation.
- The authors need to review the entire manuscript to address grammar and punctuation inconsistencies.
-Ensure that all references are the most recent and relevant to the arguments in the paper.
In summary, I recommend this manuscript for publication in Toxics after making all the corrections suggested in the reviewed version.
Reviewer 2 Report
Comments and Suggestions for Authors
Summary:
Under the background of extensive usage of chlorine-based disinfectants in Wuhan City during the COVID-19 pandemic, the author sampled drinking water from three different geological sources within Wuhan city, carried out systemic analytical characterizations of the water and experiments to explore the effects several key factors, such as reaction temperature and time, Br- and Cl- concentration, on the formation of a few DBPs compounds, and performed statistical analyses on the association of the DBPs and the measured bulk water quality indicators. While the author provided overall good writing and discussion on the data presented, and stated this manuscript is to provide theoretical support for the control of DBPs in future studies, how the presented data could be tied to the disinfection pollution event on Jan-Mar 2020 is not discussed. Below are more detailed comments.
1. In line 57:” thus guarantee the safety of public drinking water.”. should be “thus guaranteeing the safety of public drinking water.”
2. In line 58-60. “… the formation of DBPs, When…” should be “… the formation of DBPs. When…”.
3. In line 34-35 within the introduction section, the author stated over 5000 tons of chlorine-based disinfectants were used in Jan-Mar 2020. In section 2.2, while the author provided detailed sampling location map, the actual sampling time is not provided. Given the background provided by the author, it’s important to know the time gap between Jan-Mar 2020 and the actual sampling; Additionally, is this just a one-time sampling study or a follow-up study where multiple samples over time were acquired? And is there any control sample, e.g, pre-pandemic water sample, that’s available? These will better help understand the implication of the extensive usage of the chlorine-based disinfectants during the outbreak of covid.
4. In line 130. “it was subjected to agitate at 100 rpm…” should be “it was subjected to agitation at 100 rpm…”
5. For results in 3.1, the author presented one-way ANOVA results on sampling methods. From method section 2.2, the author sampled water at two depth levels: 10% and 25%, and using two types of collection bottles, glass and polyethylene. The entire discussion in 3.1 on sampling methods is around sampling bottle type, no discussion on sampling depth level is found within section 3.1 and the remaining results discussion. Thus, it is important for the author to add relevant data or discussion to show whether sampling depth is a significant factor as well.
6. In line 226-229, the author used Cb, Hb, Lb for samples in glass bottles from the Yangtze River, Han River, and the Liangzi Lake, while Cs, Hs and Ls for samples in polyethylene bottles from the Yangtze River, Han River, and the Liangzi Lake. However, the Figure 2b legend and line 101-104 used LG, HG, and CG for glass bottle samples and LP, HP, CP for polyethylene bottle samples. Thus, it is important to keep consistent on sample naming.
7. In line 257-258. The author stated “Overall, the TCM content in the water samples from all three sources showed an increasing trend over time.” I would recommend revise this conclusion sentence given that Figure 3 data does not show such increasing trend over time for TCM content.
8. Figure S1 appeared in both the main manuscript and the supplementary material in the pdf version. The one in the main manuscript should be deleted.
9. In line 397-398, a space should be given between “China(No. 22106041)” and “Province(No. 2021CFB069).”. Corrected as “China (No. 22106041)” and “Province (No. 2021CFB069).”

Round 2
Reviewer 1 Report
Comments and Suggestions for Authors
The authors replied to my comments and they have provided a new and improved version of the paper.
Reviewer 2 Report
Comments and Suggestions for Authors
Thanks for the author to have addressed all my questions.